# A Comparison of the Conventional PiG Marker Method Versus a Cluster-Based Model when recording Gait Kinematics in Trans-Tibial Prosthesis Users and the Implications for Future IMU Gait Analysis

**DOI:** 10.3390/s20051255

**Published:** 2020-02-25

**Authors:** Manunchaya Samala, Philip Rowe, Jutima Rattanakoch, Gary Guerra

**Affiliations:** 1Sirindhorn School of Prosthetics and Orthotics, Faculty of Medicine, Siriraj Hospital, Mahidol University, Bangkok 10700, Thailand; jutima.rat@mahidol.edu (J.R.); gary.gue@mahdiol.edu (G.G.); 2Department of Biomedical Engineering, University of Strathclyde, Glasgow G4 0LN, UK; philip.rowe@strath.ac.uk

**Keywords:** prosthetics, motion analysis, amputees, variability, optical tracking, inertial measurement units

## Abstract

Validation testing is a necessary step for inertial measurement unit (IMU) motion analysis for research and clinical use. Optical tracking systems utilize marker models which must be precise in measurement and mitigate skin artifacts. Prosthesis wearers present challenges to optical tracking marker model choice. Seven participants were recruited and underwent simultaneous motion capture from two marker sets; Plug in Gait (PiG) and the Strathclyde Cluster Model (SCM). Variability of joint kinematics within and between subjects was evaluated. Variability was higher for PiG than SCM for all parameters. The within-subjects variability as reported by the average standard deviation (SD), was below 5.6° for all rotations of the hip on the prosthesis side for all participants for both methods, with an average of 2.1° for PiG and 2.5° for SCM. Statistically significant differences in joint parameters caused by a change in the protocol were evident in the sagittal plane (*p* < 0.05) on the amputated side. Trans-tibial gait analysis was best achieved by use of the SCM. The SCM protocol appeared to provide kinematic measurements with a smaller variability than that of the PiG. Validation studies for prosthesis wearer populations must reconsider the marker protocol for gold standard comparisons with IMUs.

## 1. Introduction

Optoelectronic motion capture systems (OMC) have provided biomechanics professionals a tool to explain the how and why of locomotion [1,2,3]. These OMC systems have been well established as gold standard methods of quantifying joint kinematics and kinetics [4]. Another achievement in motion analysis has been the development of inertial measurement units (IMUs). These systems have ushered in a new era in human motion analysis. There are many examples in the literature of single and multi-IMU systems being utilized for the analysis of posture [5,6,7], walking [8]. These systems offer an advantage over optical motion analysis systems because of their portability. The need for research which seeks to answer questions related to able-bodied and limb different populations in ‘free-living environments’ is strongly warranted. The IMUs could fill a gap in research and serve as a clinician friendly outcome measurement instrument for prosthetist and orthotists in developed and even resource-limited environments, however, the validation of such systems against gold-standard optical tracking motion analysis systems in persons with prosthesis is still needed.

Several scholars have performed validation and reliability testing of IMU systems compared to optical systems. Validation and reliability testing is a necessary step if these IMUs are to be used for research or clinical decision making purposes. Typically, the systems collect IMUs joint kinematic metrics whilst also collecting marker based optical system joint kinematics. The IMUs are typically placed on the participants body at anatomical landmarks or along the lumbar back, left and right thigh and shanks, as well as midfoot of feet. In contrast, the opto-reflective marker sets utilized for validation studies have typically used the Plug in Gait (PiG) (Vicon Peak^®^, Oxford, UK) marker set [9,10]. The PiG is a widely utilized marker set in both able-bodied motion analysis [11,12], as well as prosthesis wearer scholarship [13,14,15].

However, the PiG is still the most common Conventional Gait Model (CGM) used, and most certainly the most minimalistic. The ease and speed of preparing a participant is worthy especially for clinical gait analysis. Two caveats of using the PiG is that even small misplacements of markers on joint centers can lead to possible errors, and tracking might be limited to three degrees-of-freedom (3DoF). Some research has questioned the validity as well as reliability of using the PiG [16,17]. The cluster-based marker (CBM) set allows six degrees-of-freedom (6DoF) and thus, independent body segments analysis at a cost of increased preparation and data analysis time [18,19]. Moreover, incorrectly placed markers and skin artifacts is a well-known issue which has previously been explored by test–retest and intra-tester reliability studies [19,20,21]. Although the PiG is common for clinical use [22], setting up markers in clusters might streamline analysis. Cluster-based markers can evaluate higher degrees-of-freedom, and although not critical to evaluate in IMUs, restriction to three degrees-of-freedom (3DoF) can alter kinematic results [23].

Prosthesis technologies have seen marked advances in materials and designs [24,25]. Thus, clinical evaluation of new types of prosthesis will merit quantitative outcome measurement of devices. The scholarship evaluating IMU accuracy with optical systems in prosthesis wearers is in its infancy, and as this scholarship will likely employ optical systems as a criterion measure, it is essential that the marker sets utilized in these studies are robust and reliable [21]. If IMUs are to eventually be used in free-living settings such as the prosthetic clinics, then the ground work in determining the most appropriate model for prosthesis wearers must be performed. Therein lies the kernel objective of this study, which was to evaluate the consistency of the Strathclyde Cluster Model (SCM) against the PiG in order to identify suitability for trans-tibial prosthesis motion analysis in IMU validation studies.

## 2. Materials and Methods

### 2.1. Participants

This study was approved by the universities Institutional Review Board (IRB) and written informed consent was obtained from all participants before study commencement. A convenience sample of seven trans-tibial amputee participants (5 male, 2 female) ages 55 ± 8 years, body weight 67 ± 6.6 kg, and height 1.73 ± 0.05 m were recruited for the study. All participants were in good health and walking without an assistive device, and with all joint ranges of motion within normal limits as assessed by the prosthetist. Participants were permitted to use their own prosthesis for study participation and all participants were provided standard walking shoes. The study prosthetist evaluated each prosthesis and ambulation of each participant in order to ensure optimal dynamic alignment of prosthesis and walking ability.

### 2.2. Protocol

Simultaneous data capture from two marker sets was accomplished by designing a marker-set which combined protocols of the PiG and the SCM, in Figure 1. This combination of marker sets is not typical, however, streamlined our data collection across our amputee participants [26]. One investigator applied both marker sets on all study participants. Static calibration and calibration techniques guided location placement of clusters. Calibration was performed for all participants after which they were then asked to walk at their self-selected speed along a 10-m indoor walkway. The participants were permitted to walk along the walkway prior to data collection after which three successful gait cycles were collected for kinematic analysis. A 10 camera VICON^®^ (VICON MX Giganet, Oxford Metrics Ltd., Oxford, UK) infrared motion capture system sampling at 100 samples per second was used to derive three-dimensional kinematics of the lower limbs. A floor mounted AMTI force plate (AMTI, Watertown, MA, USA) with data sampled at 1000 Hz was used to derive kinetics. A foot vGRF threshold of 20 N was used to determine heel-contact, and toe-off as the last frame exceeding 20 N of the participants.

### 2.3. Data Processing

After completion of data collection, the kinematic trial data was processed using VICON’s Polygon^®^ (VICON, Oxford Metrics Ltd., Oxford, UK). Reconstruction, labeling of 3D marker trajectories and gap filling was performed for both marker sets. Force plate data and the auto correlates event in the software pipeline was used to determine heel contact on the force plate [27]. Time normalized gait cycle data for both limbs were extracted from each of the participant’s three successful trials. We acknowledge the use of more trials is common, however, our participants restricted mobility resulted in our decision to limit trials. All data were filtered using a second-order Butterworth filter with a cut-off frequency of 6 Hz. The marker trajectory data was processed and gait data calculated separately for each data set according to the gait analysis software package for that marker model. The mean values of joint rotation were determined (Table 1, Table 2, Table 3, Table 4, Table 5 and Table 6). A paired t-test with a significance level of α = 0.05 was used to evaluate differences between PiG and SCM marker sets. In addition, key kinematic parameters were compared using Bonferroni corrected t-tests (α = 0.005). To observe variability, standard deviation SD and the average of the standard deviation throughout the gait cycle were analyzed. Within-subject variability was obtained by looking at three repetitions of the gait cycle. The SD was calculated for each percentage point of the gait cycle and then averaged over the cycle to give the average SD within that subject for that specific kinematic variable. Between-subject variability was defined by taking the mean cycle for each participant and calculating the within-group SD. This was then averaged across the gait cycle to provide the between-subject variability [28].

## 3. Results

To allow an overall analysis of within-subject variability, repeatability of joint angle calculation for each participant was summarized by evaluating the SD in degrees of each joint angle throughout the gait cycle. These values are reported in Table 1 and Table 2 for hip rotations, in Table 3 and Table 4 for knee rotations, and Table 5 and Table 6 for ankle rotations. In Table 1, the within-subjects variability as reported by the average SD, was below 5.6° for all rotations of the hip on the prosthesis side for all participants for both methods, with an average of 2.1° for PiG and 2.5° for SCM. Table 2, which shows the same data for the sound hip, a maximum value of 4.7° was observed and an average value of 1.9° and 2.1° for PiG and SCM, respectively. Overall, good within-subject repeatability was observed among the measured variables for both methods in the assessment of hip joint kinematics on both sides.

Table 3 illustrates the average SD obtained for the knee joint on the amputated side. Average standard deviation values, were limited to 6.5° for all knee rotations on the amputated side using the SCM and to 9.8° for PiG. Table 4 shows the average SD obtained for the sound knee with average standard deviation values being limited to 7.3° for all knee rotations on the sound side using the SCM and to 4.1° for PiG. For the PiG, the average SD for all knee joint kinematics on the amputated side was 2.5°, while for SCM, it was 2.7° (Table 3). On the sound side, average SD were 1.9° for PiG and 2.4° for SCM (Table 4).

Average SD obtained for the ankle joint on the amputated side, as determined from markers on the shoe are provided in Table 5. These values were limited to 2.9° for all ankle rotations on the amputated side for SCM and to 15.7° for PiG. Table 6 shows the average SD obtained for the ankle joint on the sound side, these average SD values for the sound side ankle were limited to 14.2° for SCM and to 8.2° for PiG. The average SD for ankle kinematics on the amputated side was 3.3° for PiG and 1° for the SCM, whereas on the sound side the PiG, was 3.1° and 3.4° for the SCM. Overall, across all three joints, the average SD for within-subjects variability was 2.5° for the PiG and 2.4° for the SCM. Both protocols showed good within-subject repeatability across all degrees of freedom for all joints.

Key kinematic parameters (Table 7, Table 8 and Table 9) were investigated and a paired t-test with a significance level of α = 0. 05 was used to test the difference between PiG and SCM protocol over the gait cycle. Statistically significant differences in joint parameters caused by a change in the protocol were evident in the sagittal plane (*p* < 0.05) on the amputated side (AH1: hip flexion/extension and AH3: peak swing flexion). For hip rotation on the sound side, there was a significant difference between AH3 and AH5, however, all other parameters showed no significant differences. For the knee, SCM estimated a higher angle for sagittal plane joint angle parameters (AK1, AK2, AK3, SK1, SK2, and SK3). The variability was higher for PiG than SCM for all parameters. Statistically significant differences in joint parameters with a change in the protocol were evident in almost all parameters on the amputated side (*p* < 0.05), except the knee parameter in the coronal plane (AK4) where no significant differences between the two protocols were found. All transverse and coronal plane parameters extracted from the gait on the amputated side, AK3, AK4, and AK5 were significantly different between PiG and SCM (*p* < 0.05). Differences in stance dorsiflexion on the amputated side were significant (*p* < 0.01). Moreover, variability was lower for SCM in all parameters.

Between-subject variability for the two limbs across nine kinematic variables is shown in Table 10 and Table 11. The between-subject variability ranged on the amputated side from 3.8° to 40.7° for PiG with an average of 17.6°, and from 3.5° to 22.6° for SCM with an average of 9.3°. For the sound side, PiG ranged from 5.7° to 22.6° with an average of 15.4°, while the SCM ranged from 3.8° to 19.3° with an average of 9.7°. The mean standard deviation for the PiG was larger than SCM on both amputated and sound limb. The overall between-subject variability of the SCM was similar on both the amputated limb 9.3° and sound limb 9.7°.

## 4. Discussion

This study aimed to compare kinematic output of two popular marker protocols in gait of seven trans-tibial amputees. A single marker set created using PiG and SCM protocols was used for this research project. The two protocols differed in degrees of assigned freedom, data processing, and calculated results. Plug in Gait required both, the attachment of a single reflective markers on the skin at anatomical landmarks as well as anthropometric measurements. The Strathclyde Cluster protocol, required clusters to be attached to limb segments, with a single reflective marker being required for anatomical calibration.

Both protocols behaved well in terms of within-subject variability, although slightly higher values of standard deviation were observed in the SCM protocol. A within-subject variability for kinematic results was observed and ranged from 0.80 to 4.60 across the two protocols. Although slightly higher values of standard deviation were observed in the SCM protocol, this was considered reasonable and did not exceed 5° [29]. Between-subject variability was higher for the PiG protocol than for the SCM protocol in most parameters.

The most remarkable result of the current study was the high standard deviation observed for the PiG protocol, particularly in the coronal and transverse planes. In contrast, the SCM showed the lowest between-subject variability. The fact that the SCM model utilized a rigid cluster of markers firmly attached to segments might have mitigated skin movement artefacts. This echoes previous research which noted that reduced errors in light of soft tissue movement when using cluster-based protocol [30,31]. The PiG is a skin-marker model SMS which requires attaching markers to anatomical landmarks. In trans-tibial amputees, markers cannot attach directly to the skin either because it has been amputated or because it is covered by a prosthesis socket. It is, therefore, necessary to attach markers to the outside of the prosthetic socket which covers the anatomical lateral epicondyles. As there is a degree of relative movement between the prosthetic socket and the residual limb, the calculation of joint kinematics does not necessarily reflect the true motion, and this presents an additional source of error [32]. The PiG showed the largest variability in most gait parameters explored. The use of skin markers can lead to a considerable high level of error [33,34]. Moreover, previous research has suggested that the placement of markers on bony landmarks should be avoided where possible as this can lead to high levels of error due to soft tissue artifacts [33].

From kinematic result comparisons of both protocols, it is clear that in our study, gait analysis was best achieved by use of the SCM. The SCM protocol appeared to provide kinematic measurements with a smaller variability than that of the PiG. The SCM also has some practical advantages over the PiG for prosthesis wearers, such as the ability to use a cluster of markers on the prosthesis which can enhance accuracy of measurement. This study is not without its limitations as our convenient sample of seven participants was limited in size, amputation level, and had a range of prostheses. Our gait analysis was restricted to laboratory settings and did not consider the variation in gait possible when walking in free-living environments. In addition, we chose three trials for our analysis which might have had implications on our overall analysis. Moreover, the lack of reliability and validity studies exploring these selected marker models in lower limb amputees makes it difficult to make broad recommendations for using either model in this population.

Still, results from this study evidenced that SCM can be an accurate and practical protocol for measuring amputee gait compared to the common PiG protocol. This result has important implications for sensor-based motion analysis for clinical use and surgical considerations. As previously mentioned, the utilization of IMUs for motion analysis research is emerging as an effective way of gait analysis outside of the confines of the motion analysis laboratory. As such, validation studies for prosthesis wearer populations must reconsider the marker protocol for gold standard comparisons with IMUs. Despite, the tendency to use Plug in Gait, we recommend cluster-based models for inertial measurement unit validation studies.

## Figures and Tables

**Figure 1 sensors-20-01255-f001:**
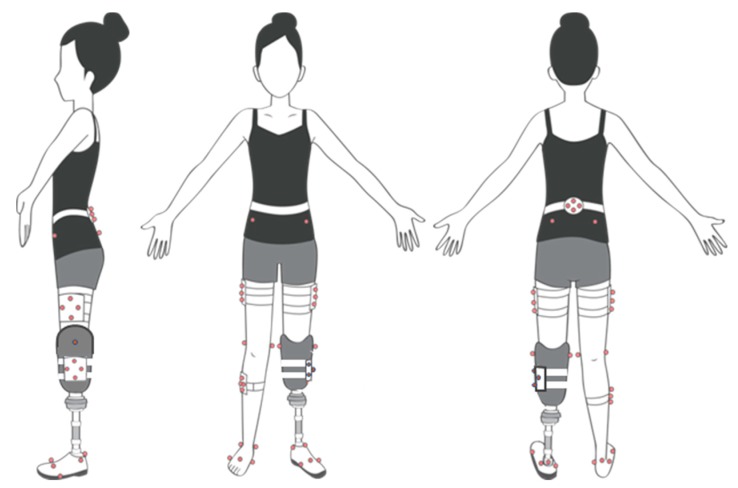
A single comprehensive marker set for two marker models Strathclyde Cluster Model (SCM) and Plug in Gait PiG).

**Table 1 sensors-20-01255-t001:** Within-subject variability of hip joint rotation over the amputee gait cycle across three trials per participant as calculated by PiG and SCM is shown by the standard deviation, SD.

Participant	Hip Joint Rotation					
	Flex/Extension		Ab/Adduction		Int/External	
	PiG	SCM	PiG	SCM	PiG	SCM
Subject 1	1.6	2.4	1.1	1.0	2.0	1.5
Subject 2	1.3	1.0	0.8	1.6	1.8	1.8
Subject 3	1.0	1.8	1.3	2.7	3.6	2.7
Subject 4	1.9	2.6	1.3	2.6	5.6	2.2
Subject 5	2.0	1.8	1.2	2.0	1.7	2.5
Subject 6	2.5	5.6	1.6	2.4	3.0	2.9
Subject 7	3.1	3.0	1.2	3.1	4.4	4.6
**Mean** (°)	1.9	2.6	1.2	2.2	3.2	2.6

**Table 2 sensors-20-01255-t002:** Within-subject variability of hip joint rotation over the sound gait cycle across three trials per participant as calculated by PiG and SCM is shown by the standard deviation, SD.

Participant	Hip Joint Rotation					
	Flex/Extension		Ab/Adduction		Int/External	
	PiG	SCM	PiG	SCM	PiG	SCM
Subject 1	1.0	1.9	0.4	0.6	1.6	1.6
Subject 2	1.3	1.3	0.6	1.1	1.3	1.8
Subject 3	3.1	4.3	0.7	2.8	2.5	1.3
Subject 4	2.0	4.1	1.1	2.7	2.9	3.9
Subject 5	3.3	3.4	1.1	1.8	2.7	2.7
Subject 6	2.5	3.9	0.8	1.8	2.7	4.3
Subject 7	2.2	2.5	1.0	1.8	4.7	1.5
**Mean** (°)	2.2	3.0	0.8	1.8	2.6	2.4

**Table 3 sensors-20-01255-t003:** Within-subject variability of knee joint rotation over the amputated gait cycle across three trials per participant as calculated by PiG and SCM is shown by the standard deviation, SD.

Participant	Knee Joint Rotation					
	Flex/Extension		Ab/Adduction		Int/External	
	PiG	SCM	PiG	SCM	PiG	SCM
Subject 1	2.0	4.5	1.8	2.7	1.0	2.2
Subject 2	2.1	1.6	1.1	1.4	0.7	1.6
Subject 3	1.1	1.8	1.6	1.4	0.3	2.3
Subject 4	2.2	3.9	3.2	2.4	1.4	2.2
Subject 5	2.4	3.4	2.0	1.4	2.9	2.0
Subject 6	4.9	6.5	3.4	1.7	0.7	2.6
Subject 7	5.3	3.9	3.2	2.9	9.8	4.4
**Mean** (°)	2.9	3.7	2.3	2.0	2.4	2.5

**Table 4 sensors-20-01255-t004:** Within-subject variability of knee joint rotation over the sound gait cycle across three trials per participant as calculated by PiG and SCM is shown by the standard deviation, SD.

Participant	Knee Joint Rotation					
	Flex/Extension		Ab/Adduction		Int/External	
	PiG	SCM	PiG	SCM	PiG	SCM
Subject 1	0.8	2.8	0.5	0.8	1.0	1.8
Subject 2	1.3	2.1	1.5	0.8	1.6	1.5
Subject 3	4.1	6.1	2.5	1.9	2.4	3.9
Subject 4	2.3	3.0	1.4	2.6	2.0	2.5
Subject 5	3.2	7.3	1.9	1.9	2.8	2.3
Subject 6	2.0	2.3	1.7	1.3	3.9	1.8
Subject 7	2.4	4.7	3.3	2.0	2.8	2.3
**Mean** (°)	2.3	4.1	1.8	1.6	2.4	2.3

**Table 5 sensors-20-01255-t005:** Within-subject variability of ankle joint rotation over the amputated gait cycle across three trials per participant as calculated by PiG and SCM is shown by the standard deviation, SD.

Participant	Ankle Joint Rotation SD					
	Dorsi/Plantar		Ab/Adduction		Inv/Eversion	
	PiG	SCM	PiG	SCM	PiG	SCM
Subject 1	1.1	0.9	1.2	2.9	5.5	1.4
Subject 2	0.3	0.5	0.2	0.8	1.5	0.7
Subject 3	0.5	0.5	0.9	0.8	3.6	0.8
Subject 4	1.0	0.8	3.7	1.1	2.2	0.8
Subject 5	6.4	0.6	6.9	0.7	1.7	0.6
Subject 6	1.1	1.5	0.4	1.0	2.4	1.3
Subject 7	11.1	1.6	3.3	1.2	15.7	1.4
**Mean** (°)	3.1	0.9	2.3	1.2	4.6	1.0

**Table 6 sensors-20-01255-t006:** Within-subject variability of ankle joint rotation over the sound gait cycle across three trials per participant as calculated by PiG and SCM is shown by the standard deviation, SD.

Participant	Ankle Joint Rotation SD					
	Dorsi/Plantar		Ab/Adduction		Inv/Eversion	
	PiG	SCM	PiG	SCM	PiG	SCM
Subject 1	1.7	2.4	0.9	1.9	1.3	1.1
Subject 2	6.4	1.5	3.6	1.7	1.5	1.0
Subject 3	8.2	2.8	8.0	5.6	1.1	1.4
Subject 4	1.8	2.0	0.8	2.5	3.4	1.4
Subject 5	4.1	6.0	1.5	4.1	4.5	2.5
Subject 6	1.8	14.2	1.8	12.2	4.2	1.5
Subject 7	1.8	1.7	1.1	2.5	5.0	1.3
**Mean** (°)	3.7	4.4	2.5	4.4	3.0	1.5

**Table 7 sensors-20-01255-t007:** Hip joint angle parameters of seven amputee participants on sound and amputated side as mean SD over three gait cycles calculated by the PiG and SCM.

Parameters	PiG SD	SCM SD	*p*-Value	*p* < 0.05	*p* < 0.005
**Sound side**					
Hip flex/extension ROM	38.4(4.7)	42.1(7.0)	0.12		
Peak Stance Extension	15.3(10.5)	−8.9(10.0)	0.06	*	
Peak Swing Flexion	17.9(12.4)	27.2(13.5)	0.02		
Hip Ab/Ad ROM	9.1(3.2)	12.3(3.3)	0.11	*	
Hip Int/Ext Rotation ROM	24.6(5.3)	12.1(4.0)	0.01		
**Amputated side**					
Hip flex/extension ROM	42.1(7.6)	49.0(5.6)	0.00	*	**
Peak Stance Extension	−11.4(9.6)	−7.6(9.2)	0.22		
Peak Swing Flexion	27.5(8.8)	38.5(9.7)	0.01	*	
Hip Ab/Ad ROM	8.9(3.8)	11.6(3.3)	0.32		
Hip Int/Ext Rotation ROM	37.6(32.3)	12.3(3.0)	0.09		

* Significant difference (α = 0.05), ** Indicates significance level of *p* < 0.005 after Bonferroni correction (0.05/10).

**Table 8 sensors-20-01255-t008:** Knee joint angle parameters of 7 amputee participants on sound and amputated side as mean SD over three gait cycles calculated by the PiG and SCM.

Parameters	PiG SD	SCM SD	*p*-Value	*p* < 0.05	*p* < 0.005
**Sound side**					
Knee flex/extension ROM	48.8(10.0)	63.4(7.7)	0.01	*	
Peak Stance Extension	3.2(11.4)	8.6(8.4)	0.18		
Peak Swing Flexion	43.6(14.3)	60.8(7.5)	0.02	*	
Knee Ab/Ad ROM	34.1(12.6)	21.3(9.4)	0.08		
Knee Int/Ext Rotation ROM	20.7(8.1)	19.5(5.2)	0.71		
**Amputated side**					
Knee flex/extension ROM	45.3(12.8)	70.6(8.9)	0.01	*	
Peak Stance Extension	6.8(7.2)	12.2(4.4)	0.03	*	
Peak Swing Flexion	41.9(14.6)	69.3(5.4)	0.01	*	
Knee Ab/Ad ROM	38.7(15.1)	22.5(10.0)	0.10		
Knee Int/Ext Rotation ROM	9.8(6.9)	21.2(5.6)	0.01	*	

* Significant difference (α = 0.05).

**Table 9 sensors-20-01255-t009:** Ankle joint angle parameters of seven amputee participants on sound and amputated side as mean SD over three gait cycles calculated by the PiG and SCM.

Parameters	PiG SD	SCM SD	*p*-Value	*p* < 0.05	*p* < 0.005
**Sound side**					
Ankle Plantar/dorsiflexion	37.3(16.1)	25.7(3.5)	0.12		
Peak Stance dorsiflexion	32.8(21.8)	2.8(6.6)	0.02	*	
Peak Swing plantarflexion	−2.3(8.5)	−21.1(6.6)	0.01	*	
Ankle Ab/Adduction	13.5(13.7)	11.6(3.9)	0.76		
Ankle Inv/Eversion ROM	11.5(5.0)	10.6(3.6)	0.66		
**Amputated side**					
Ankle Plantar/dorsiflexion	10.5(6.0)	8.2(2.8)	0.43		
Peak Stance dorsiflexion	15.2(11.8)	−3.0(2.6)	0.01	*	
Peak Swing plantarflexion	9.0(13.3)	−8.2(4.0)	0.02	*	
Ankle Ab/Adduction	14.4(11.5)	3.2(0.8)	0.04	*	
Ankle Inv/Eversion ROM	34.7(30.5)	4.0(1.3)	0.04	*	

* Significant difference (α = 0.05).

**Table 10 sensors-20-01255-t010:** Between-subject variability illustrated by the mean standard deviation over the amputee gait cycle among the seven participants for both protocols.

Amputated Side										
	Hip Joint			Knee Joint			Ankle Joint			Mean (°)
Protocol	Flex/Ext	Ab/Ad	In/Ex	Flex/Ext	Ab/Ad	In/Ex	Dorsi/Plntar	Ab/Ad	Inv/Evr	
PiG	9.6	3.8	38.3	10.0	11.1	12.7	14.0	18.3	40.7	17.6
SCM	10.6	4.6	22.4	8.2	9.3	15.5	3.5	5.2	4.4	9.3

**Table 11 sensors-20-01255-t011:** Between-subject variability illustrated by the mean standard deviation over the sound gait cycle among the seven participants for both protocols.

Sound Side										
	Hip Joint			Knee Joint			Ankle Joint			Mean (°)
Protocol	Flex/Ext	Ab/Ad	In/Ex	Flex/Ext	Ab/Ad	In/Ex	Dorsi/Plntar	Ab/Ad	Inv/Evr	
PiG	11.4	5.7	19.0	10.5	8.1	22.6	17.6	22.0	21.4	15.4
SCM	11.8	3.8	19.3	7.4	10.3	14.7	6.7	6.6	6.5	9.7

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
