# Peer review of "A Comparison of the Conventional PiG Marker Method Versus a Cluster-Based Model when recording Gait Kinematics in Trans-Tibial Prosthesis Users and the Implications for Future IMU Gait Analysis"

_sensors, 2020, doi:10.3390/s20051255_

Round 1
Reviewer 1 Report
Conventional gait versus cluster model kinematics in trans-tibial prothesis users: Implications for IMU gait analysis
Heading
Line 3: Prothesis: respelling
Line 2-4: Consider changing to a clearer headline which also includes skin marker model such as "Conventional skin marker model versus cluster marker model kinematics in transtibial prosthesis users: Implications for validation of IMU devices"
Abstract
Line 17 (PiG) and the Strathclyde Cluster Model (SCM): Print the “Plug in Gate” with the abbreviation in brackets. See line 52. Then use the abbreviation without parentheses throughout the article.
Line 22-23 (AH1 and AH3): Try not use abbreviations without explanation in running text
Introduktion
Line 36: Consider rewrite sentence since they have been in use for some time, consider also the next following sentence. Consider also reducing the number of IMU references in favour of reliability and validity studies conducted on PiG and SCM models.
Line 44: Please strike (RLE) when it does not return
Line 46-55: Do the referenced articles in this section are related to reliability and validity studies with PiG and SCM marker models? Please clarify and add.
Please review the manuscript and the numbers and parentheses on references according to the journal's rules.
Line: 56-57: Is it possible to quantify which is the most common marker model at all and which is the easiest gait analysis marker model? Please clarify and refer.
Line 58: Placing markers incorrectly is a well-known artefact together with soft tissue artefacts which can be minimized. Here one can refer to test retest studies of marker placement in order to highlight the problem. Please refer to Pig and SCM used marker models.
Line 59-61: Is it necessary to use systems based on 6 DoF principles to evaluate IMUs. Please clarify.
Line 64 “robust and reliable”: Please include some references to reliable and validated studies on the marker models used in the study.
Line 67: Consider using the same abbreviation SCM without parentheses throughout the manuscript. Try to avoid adding new abbreviations if not necessary.
2 Materials and Methods
2.1.
Please add descriptive information about the seven study participants.
2.2.
Line 80: Please use PiG and SCM throughout the article without parentheses. CMB is a new acronym that needs to be printed out with the acronym parentheses. If the word is not used again remove the abbreviation completely.
Consider removing "which is a CMB model, as shown in".
Line 79-101 with Figure 1: The used marker models are modified and adapted to amputees in different ways due to the position of the markers on the amputated sleeve, which means that it can be difficult to refer to other studies that have used and conducted reliability and validity studies with the PiG and SCM marker models. Please rewrite and clarify.
Furthermore, please clarify if the subjects used shoes. If so, it also entails a modification of the marker models and another source of error. Please clarify.
Figure 1: Moreover, the location of the cluster on the back appears to be located in the lumbar region and not on the pelvis. This position makes it rather difficult, if it is even possible, to calculate the hip joint movement in all planes. Please clarify.
Line 105: Please clarify how heel contact and too off was recognized.
Line 109 "three best trials": How many trials, in total, did each subject perform? Please clarify why you chose the 3 best trials and not all and/or not randomized among all trials.
Line 112: Please rewrite line 112
Line 113 ”key kinematic parameters”: Please clarify what is meant and refer to Table 7-9
Results
Line 124: Please write SD in parentheses and remove parentheses around "in degrees". Please see line 116. Then you can use SD throughout the article and in tables without parentheses.
Line 163: How to handle the prosthetic ankle in a shoe and how to interpret the result? Please develop and clarify
Line 180: Please clarify what meant by AH1 and AH3.
Line 204: Please change the degree sign.
The study has a number of limitations such as few study participants, lack of reliability and validity studies in lower leg amputated with used marker models, which means that one should be more careful about the conclusion. Consider revising the conclusion.
Author Response
Heading
Line 3: Prothesis: respelling
Response: Prosthesis, corrected.
Line 2-4: Consider changing to a clearer headline which also includes skin marker model such as "Conventional skin marker model versus cluster marker model kinematics in transtibial prosthesis users: Implications for validation of IMU devices"
Response: We have adjusted the title to reflect your consideration. “A comparison of the Conventional PiG marker method versus a cluster based model when recording gait kinematics in trans-tibial prosthesis users and the Implications for future IMU gait analysis” if this is too long we can change to “A comparison of the Conventional PiG marker method versus a cluster based model when recording gait kinematics in trans-tibial prosthesis users”
Abstract
Line 17 (PiG) and the Strathclyde Cluster Model (SCM): Print the “Plug in Gate” with the abbreviation in brackets. See line 52. Then use the abbreviation without parentheses throughout the article.
Response: We have scanned the paper through and have made adjustments to abbreviations.
Line 22-23 (AH1 and AH3): Try not use abbreviations without explanation in running text
Response: We have omitted this abbreviation so as to not confuse readers
Introduktion
Line 36: Consider rewrite sentence since they have been in use for some time, consider also the next following sentence. Consider also reducing the number of IMU references in favour of reliability and validity studies conducted on PiG and SCM models.
Response: We have removed this sentence. We have reduced IMU references in favor of reliability and validity of PiG and SCM studies (Line 79,85).
Line 44: Please strike (RLE) when it does not return
Response: We have stricken RLE.
Line 46-55: Do the referenced articles in this section are related to reliability and validity studies with PiG and SCM marker models? Please clarify and add.
Response: We have reduced references and added references referring to reliability testing of PiG and evaluation of SCM marker models. Lines 84-86.
Please review the manuscript and the numbers and parentheses on references according to the journal's rules.
Response: We have reviewed the journals rules regarding numbers and parentheses and complied.
Line: 56-57: Is it possible to quantify which is the most common marker model at all and which is the easiest gait analysis marker model? Please clarify and refer.
Response: We have addressed this point and have clarified that although the PiG is common in clinical settings, clusters make for a faster set up time. Line 86.
Line 58: Placing markers incorrectly is a well-known artefact together with soft tissue artefacts which can be minimized. Here one can refer to test retest studies of marker placement in order to highlight the problem. Please refer to Pig and SCM used marker models.
Response: We appreciate the comment and have referred and cited research which explored the test-retest and interrater reliability of motion analysis.
Line 59-61: Is it necessary to use systems based on 6 DoF principles to evaluate IMUs. Please clarify.
Response: We appreciate the comment and have added clarification in our paper. Cluster based markers can evaluate higher degrees-of-freedom, and although not critical to evaluate in IMUs, restriction to three degrees-of-freedom (3DoF) can alter kinematic results Line 90.
Line 64 “robust and reliable”: Please include some references to reliable and validated studies on the marker models used in the study.
Response: We agree with your comment and have added a reference of previous work exemplifying reliability and validity of marker models in the study. Line 93
Line 67: Consider using the same abbreviation SCM without parentheses throughout the manuscript. Try to avoid adding new abbreviations if not necessary.
Response: We have decided to stick with SCM throughout our revised manuscript.
2 Materials and Methods
2.1.
Please add descriptive information about the seven study participants.
Response: We have provided descriptive information for these participants
2.2.
Line 80: Please use PiG and SCM throughout the article without parentheses. CMB is a new acronym that needs to be printed out with the acronym parentheses. If the word is not used again remove the abbreviation completely. Consider removing "which is a CMB model, as shown in".
Response: We have changed all to PiG and SCM and have removed CMB
Line 79-101 with Figure 1: The used marker models are modified and adapted to amputees in different ways due to the position of the markers on the amputated sleeve, which means that it can be difficult to refer to other studies that have used and conducted reliability and validity studies with the PiG and SCM marker models. Please rewrite and clarify.
Furthermore, please clarify if the subjects used shoes. If so, it also entails a modification of the marker models and another source of error. Please clarify.
Response: We have rewritten to make note of these two issues in hopes of clarifying our protocol and fact of participants wearing shoes for the reader and have also added these points as limitations in our discussion section. Line 113, 106.
Figure 1: Moreover, the location of the cluster on the back appears to be located in the lumbar region and not on the pelvis. This position makes it rather difficult, if it is even possible, to calculate the hip joint movement in all planes. Please clarify.
Response: We acknowledge this and have clarified our calibration technique in the manuscript which refers to our prior scholarship using the Strathclyde Cluster Model, where we provide a diagram.
Line 105: Please clarify how heel contact and too off was recognized.
Response: Great question, we have clarified this in the manuscript. A foot vGRFthreshold of 20N was usedto determine heel-contact, and a toe-off vGRF thresholddropping below10 N was used to determine heel-contact and toe-off of the participants. Line 176
Line 109 "three best trials": How many trials, in total, did each subject perform? Please clarify why you chose the 3 best trials and not all and/or not randomized among all trials.
Response: Participants performed at least five trials but because the amputee patients had trouble with clean force strikes we choose their three best trials. We acknowledge that a greater number of trials might reduce variability in kinematic variables and also a limitation of our study to not randomizing selection of trials. This is mentioned in Line 196 and also as a limitation in our discussion, Line 738.
Line 112: Please rewrite line 112
Response: Rewritten
Line 113 ”key kinematic parameters”: Please clarify what is meant and refer to Table 7-9
Response: We have clarified hip, knee and ankle joint rotations and referred to the tables. Line 202
Results
Line 124: Please write SD in parentheses and remove parentheses around "in degrees". Please see line 116. Then you can use SD throughout the article and in tables without parentheses.
Response: Thank you, we have modified according to your suggestion and removed parentheses through ought the paper for SD.
Line 163: How to handle the prosthetic ankle in a shoe and how to interpret the result? Please develop and clarify
Response: We noted and specified markers placed on shoe for the prosthesis side. Line 376
Line 180: Please clarify what meant by AH1 and AH3.
Response: These are now clarified in the manuscript as AH1:Hip flexion/extension ROM at amputated side, AH3 Peak swing flexion at amputated side
Line 204: Please change the degree sign.
Response: We have adjusted
The study has a number of limitations such as few study participants, lack of reliability and validity studies in lower leg amputated with used marker models, which means that one should be more careful about the conclusion. Consider revising the conclusion.
Response: We agree with your comments and have adjusted our conclusions as well as highlighted these limitations, Line 699
Reviewer 2 Report
The paper is well structured and the treated topic falls into the SENSORS aim and scope.
However some minor concerns should be addressed by the Authors in order to have a final more strong paper.
Authors should underline the limitation of the value of the study, and the clinical and surgical implication of the presented study should be added. At this stage the paper seems to be directed to surgeons and not researchers. Please emphasize the clinical application of the study, and its scientific rationale.
Some recent references about Prosthesis procedure and clinical implication should be added for increasing the introduction section
Fiorillo, L.; D’Amico, C.; Turkina, A.Y.; Nicita, F.; Amoroso, G.; Risitano, G. Endo and Exoskeleton: New Technologies on Composite Materials. Prosthesis 2020, 2, 1-9
Cicciù, M. Prosthesis: New Technological Opportunities and Innovative Biomedical Devices. Prosthesis 2019, 1, 1-2
Author Response
The paper is well structured and the treated topic falls into the SENSORS aim and scope.
However some minor concerns should be addressed by the Authors in order to have a final more strong paper.
Authors should underline the limitation of the value of the study, and the clinical and surgical implication of the presented study should be added. At this stage the paper seems to be directed to surgeons and not researchers. Please emphasize the clinical application of the study, and its scientific rationale.
Response: Thank you for the comments, we have stated our limitations in the value of the study Line 701. We have noted the clinical and surgical implication of the present study as well Line 706.
Some recent references about Prosthesis procedure and clinical implication should be added for increasing the introduction section
Fiorillo, L.; D’Amico, C.; Turkina, A.Y.; Nicita, F.; Amoroso, G.; Risitano, G. Endo and Exoskeleton: New Technologies on Composite Materials. Prosthesis 2020, 2, 1-9
Cicciù, M. Prosthesis: New Technological Opportunities and Innovative Biomedical Devices. Prosthesis 2019, 1, 1-2
Response: Thank you for the references, we have made note of this information and cited our paper accordingly. Line 91